# Synthesis, Characterization, and Biological Evaluation of 2-(N-((2′-(2H-tetrazole-5-yl)-[1,1′-biphenyl]-4yl)-methyl)-pentanamido)-3-methyl Butanoic Acid Derivatives

**DOI:** 10.3390/molecules28041908

**Published:** 2023-02-16

**Authors:** Anum Masood, Mohsin Abbas Khan, Irshad Ahmad, Asim Raza, Farhat Ullah, Syed Adnan Ali Shah

**Affiliations:** 1Department of Pharmaceutical Chemistry, Faculty of Pharmacy, The Islamia University of Bahawalpur, Bahawalpur 63100, Pakistan; 2Faculty of Pharmacy, Universiti Teknologi, MARA Cawangan Selangor Kampus Puncak Alam, Bandar Puncak Alam 42300, Selangor, Malaysia

**Keywords:** DPPH, FTIR, NMR, antihypertensive activity, antioxidant activity, valsartan

## Abstract

This study aimed to evaluate 2-(N-((2′-(2H-tetrazole-5-yl)-[1,1′-biphenyl]-4yl)-methyl)-pentanamido)-3-methyl butanoic acid-based ester derivatives as a new class of angiotensin-II receptor antagonists. For this purpose, a series of compounds were synthesized using a variety of phenols. Their chemical characterization was established by FTIR, ^1^HNMR, and ^13^CNMR techniques. The biological activities including antioxidant potentials using the DPPH assay, the antihypertensive assay, the urease enzyme inhibition assay, and the antibacterial assay using agar well diffusion methods were performed. All the new compounds showed significant free radical scavenging potentials more than the parent drug while retaining antihypertensive potentials along with urease inhibition properties. However, the AV2 test compound was found to be the most potent against hypertension. Most of the synthesized analogs showed urease inhibitory actions. Molecular docking studies were performed for all the active analogs to decode the binding detail of the ligands with receptors of the enzyme’s active site.

## 1. Introduction

There is a variety of compound classes to prevent, regulate and treat hypertension known as antihypertensive drugs. They may differ in their chemistry and mode of action. These drugs may also find their use in other conditions i.e., beta-blockers in anxiety, thyrotoxicosis and heart failure [1]. Ester and water are synthesized when alcohols/phenols undergo a reaction with carboxylic acids known as esterification which is a rescindable reaction thus esters may go through hydrolysis to yield alcohol along with organic acid, respectively [2]. Esterification is well known synthetic technique and plays an important role in the medicinal field. A prodrug having an ester linkage might have the potential to carry two active drugs and can work against two diseases. It is reported that paracetamol and ibuprofen can be esterified into a prodrug that can work as an antipyretic, anti-inflammatory, and analgesic [3]. Hypertension is a worldwide problem and one of the major causes of stroke which may lead to a heart attack. The renin-angiotensin system plays a significant role in blood pressure control via the functions of angiotensin II which includes the release of nor-epinephrine, the secretion of aldosterone, the re-absorption of renal sodium, and vasoconstriction. Therefore, it can be considered a beneficial intervention in handling hypertension [4]. Among the hypertensive group of drugs, valsartan (2-(N-((2′-(2H-tetrazole-5-yl)-[1,1′-biphenyl]-4yl)-methyl)-pentanamido)-3-methyl butanoic acid) is one of the most commonly used drug moieties used in the treatment of hypertension [5]. The ester derivatives of Valsartan were synthesized and assessed for their antihypertensive, urease enzyme inhibitory and antioxidant potentials. Various techniques have been used to evaluate the antioxidant potential of drugs of interest for both in vitro and in vivo studies [6]. The in vivo antihypertensive activity was performed for the synthesized ester derivatives. Research studies show that many antihypertensive drugs are modified to treat hypertension as well as hypertension-induced cardiovascular damage [7]. Urease, an enzyme belonging to the amidohydrolases family, has active sites that contain two nickel atoms. The significance of urease in enzymology is due to James B. Sumner (1926), who purified it for the first time. It was the first enzyme that was crystallized [8]. It is distributed in nature from prokaryotes to eukaryotes having a minor difference in composition, while the role is the same [9]. Urea, being a stable molecule, requires an enzyme that plays a role as a catalyst for hydrolysis to carbonic acid and ammonia [10]. However, ammonia produced as a result of urea hydrolysis leads to a rise in pH and causes alkalinity. The raised alkalinity helps in the multiplication of a pathogenic microorganism *Proteus mirabilis* and *Helicobacter pylori* resulting in infectious disorders including urolithiasis [11], ulcers [12,13], urinary tract infections (UTIs) [14], tuberculosis [15], and yersiniosis [16]. The DPPH method (α,α-diphenyl-β-picrylhydrazyl) suggests a pioneering approach for testing the antioxidant activity of desired compound [17]. Antioxidants are the compounds that can protect cells against damage caused by an entity called free radicals [18]. Keeping in view, a balance between free radicals and antioxidants is greatly required for the normal physiological functions of the human body. Research studies have intensified the search for effective and non-toxic antioxidant substances in the recent few years [19]. Research shows that high blood pressure may lead to a bacterial infection. A chronic infection (viruses and bacteria) has shown a close relation with hypertension. Moreover, hypertension also confirmed its association with the severity and mortality of SARS-CoV-2 infections in recent studies. Furthermore, hypertension plays a crucial role in the severe pathogenesis of tuberculosis [20]. The synthesized compounds (AV1-AV11) were tested for their antimicrobial potentials, and antihypertensive activity was also tested for these compounds by in vivo and in vitro models.

## 2. Results and Discussion

All the compounds showed physicochemical properties within an acceptable range. All of the derivatives showed low gastrointestinal absorption except one that is AV9, which showed high gastrointestinal absorption and zero Lipinski violation comparable with the parent molecule while not a single derivative crossed the blood-brain barrier (BBB) including the parent drug (Table 1).

Diphenyl picryl hydrazyl (DPPH) is a firm radical and can be easily employed for estimating the antioxidant potential of chemicals. All the synthesized derivatives showed higher free radical scavenging potential comparable with the standard Ascorbic acid except AV7 which showed a slightly lower potential than parent drug (AV0). Antioxidants may be responsible for preventing a wide variety of infectious diseases against pathological consequences [21]. Chemical moieties which stabilize and prevent cell damage by providing free electrons to the damaged cell are called antioxidant substances [22]. These compounds may have the potential to be used for hypertension and hypertension-induced free radical-related cardiovascular damage. All the tested compounds showed significantly higher free radical scavenging activity in a trend AV2 > AV11, AV8 > AV10 > AV3 > AV4 > AV9 > AV6 > AV1, AV5 > AV0 > AV7 (Table 2).

Angiotensin II blockers (ARBs) efficiently impart their role in lowering blood pressure with fewer adverse effects [23]. The antihypertensive activity was performed for the synthesized ester derivatives of 2-(N-((2′-(2H-tetrazole-5-yl)-[1,1′-biphenyl]-4yl)-methyl)-pentanamido)-3-methyl butanoic acid. DMSO was used for dissolving derivatives, as it has a negligible effect on blood pressure [24]. All of the test derivatives exhibited positive results as compared to the parent molecule while AV3 and AV9 showed significant results in lowering blood pressure (Table 3 and Table 4).

Initially, the synthesized derivatives were tested against urease at a concentration of 0.1 mM, and a %inhibition was calculated. The IC_50_ values of newly synthesized derivatives were calculated (Table 5). All the compounds showed different inhibition capacities. In addition, the overall trend for urease inhibition within the series was AV2 > AV5 > AV3 > AV10, AV6, AV11, AV9, AV1 > AV4, AV8 > AV7 (Table 5).

The results of the SAR-based study showed that the types of groups connected to aryl groups and their positions were key factors in determining the activity of compounds to inhibit urease [25]. Only synthetic derivatives with different substituents connected to the phenyl portion of the compounds were used to establish SAR. AV2 was found to be the most potent member among the series bearing the methoxy group at C-3′ and the aldehyde group at C-1′ with IC_50_ = 0.28 µM ± 0.15, 4 folds higher than the standard Thiourea (IC_50_ = 4.24µM ± 0.13). Another compound AV5 with IC_50_ = 1.29 µM ± 0.12 bearing chlorine at C’-2 and C’-4 exhibited 3 folds more potential of inhibition than standard thiourea. Furthermore, AV3 and AV8 (IC_50_ = 1.56 ± 0.12 and IC_50_ = 1.52 ± 0.12) having substitutions at more carbons of the phenyl ring showed 3 folds more inhibition than Thiourea. In contrast, the compound AV7 (IC_50_ = 8.59 ± 1.12) bearing only one substitution at the phenyl ring showed decreased activity. It is evident from looking at the substitution scenario in compounds AV1–AV11 that the hydrogen bonds formed by the compounds’ strong polar groups—nitro (NO_2_), phenyl (Ph), and chlorine (Cl)—make them effective against urease. On the other hand, hydroxyl (OH), methoxy (OCH_3_), and methyl (CH_3_) molecules had lower potency. It was predicted from the overall SAR analysis of the produced compounds that the compounds with strong electron withdrawing groups substituted at the reagents’ phenyl rings (A1–A11) would be highly potent. Urease enzymes in high concentration may cause great damage to human beings by assisting *Helicobacter pylori* to keep at a low pH, which leads to ulcers of the gastric mucosa. Ultimately, these ulcers may cause gastro carcinoma [26]. Moreover, when ammonia is released continuously by overexpression of urease it may lead to other metabolic ailments and permanent damage to the gastric epithelial cells resulting in death [27]. Therefore, these active compounds can be used for the above-mentioned conditions along with hypertension by further screening and development.

The antibacterial assays were conducted on all novel compounds. Regrettably, all new compounds failed to demonstrate antibacterial potentials against the aforementioned bacterial strains belongs to gram +ve (*Bacillus pumilus* and *Staphylococcus aureus*) and gram -ve (*Escherichia coli* and *Pseudomonas aeruginosa*). All derivatives were found to be inert likewise the parent molecule (Table 6).

To investigate the fitness scores of the bioactive conformations as well as their selectivity for the urease enzyme, all produced derivatives were cropped and integrated into the binding site of the enzyme (PDB: 1E9Y). By using the determined X-Ray structure of *Helicobacter pylori* urease in complex with AHA, the docking consistency had been approved. After being taken out of the composite previously stated, the ligand (AHA) was then re-docked into the binding pocket of the active site of *Helicobacter pylori* urease (Figure 1). All derivatives were analyzed; however, only two were active against urease, as shown by 1E9Y (Table 7).

One of the derivatives chemically interacted with the bi-nickel centre of the urease enzyme, according to the molecular docking experiments. The derivative ASSV4 showed strong salt bridges with ARG338 and tetrazole nitrogen. Positive hydrogens were formed between HIS322 and HIS274 with the benzene ring and the tetrazole ring of nitrogen. HIE221 makes a hydrogen bond with the tetrazole ring of nitrogen. Moreover, making five salt coordinate bonds with nickel and tetrazole nitrogen. ALA169 makes a hydrophobic interaction with the oxygen of the ester bond (Figure 1).

The docking investigations demonstrate that the AV4 variant interacted with the enzyme’s bi-nickel centre. The derivative AV4 showed strong salt bridges with ARG338 and tetrazole nitrogen. Positive hydrogens were formed between HIS322 and HIS274 with the benzene ring and tetrazole ring of nitrogen. HIE221 made a hydrogen bond with the tetrazole ring of nitrogen. Moreover, it made five salt coordinate bonds with nickel and tetrazole nitrogen. ALA169 made a hydrophobic interaction with the oxygen of the ester bond. (Figure 2).

## 3. Materials and Methods

### 3.1. General

All the chemicals and reactants used in this study were of analytical grade. Methanol, Benzoin, vanillin, picric acid, 4-amino phenol, dichlorophenol, 4-hydroxy benzaldehyde, tyrosin, thymol, alpha-naphthol, and beta-naphthol were purchased from Sigma Chemicals (St. Louis, MO, USA). DPPH (2, 2-diphenyl-1-picrylhydrazyl) and salicylic acid were purchased from Merck (Darmstadt, Germany). 2-(N-((2′-(2H-tetrazole-5-yl)-[1,1′-biphenyl]-4yl)-methyl)-pentanamido)-3-methyl butanoic acid (96%, USP grade) was obtained from Highnoon laboratories Pvt. Ltd., Lahore, Pakistan. Using the Gallen Kamp melting point apparatus, melting points were recorded. 1H and 13C nuclear magnetic resonance analysis was performed. The cary630 FTIR spectrophotometer was used to generate a Fourier transform infrared (FTIR) spectrum using the KBr pellet press technique. The glassware was dried in a UN55 (Memmert) oven at 200 °C. In 1.0 mM of each reactant, the % yields of derivatives are provided.

### 3.2. Chemistry

General Procedure for the Synthesis of Valsartan Series AV1-AV11

Valsartan coded as AV0 (1 mM) was added into 30 mL methyl alcohol (analytical grade) and used as a solvent. Then the variety of alcohols/phenols (1 mM) (Figure 3) was mixed along with concentrated sulphuric acid (catalyst). For 4 h, the reaction mixture was refluxed (Figure 1). All derivatives formed were filtered and recrystallized using ethyl alcohol to get all the products (AV1-AV11) in good yields. All The compounds were subjected to drying using a vacuum oven (Thermo Scientific) for 24 h.

AV0: IUPAC Name: 2-(N-((2′-(2H-tetrazole-5-yl)-[1,1′-biphenyl]-4yl)-methyl)-pentanamido)-3-methyl butanoic acid; melting point: 116 °C; molecular formula: C_24_H_29_N_5_O_3_ and molecular weight: 435.52 gm/mol. Elemental analysis (calculated) C_24_H_29_N_5_O_3_: C, 66.18; H, 6.69; N, 15.97; (found) C, 66.25; H, 6.65; N, 16.11, FT-IR ν(cm^−1^), 3056, 2963, 2873 (CH), 1727 (C=O), 1599, 1459 (C=C), 1129 (CN), 3252 (NH), 3539 (OH), ^1^H NMR (DMSO, ppm) δ: 0.86–0.88 (3H, t), 0.92–0.94 (6H, d) 1.17–1.19 (2H, m), 1.41–1.42 (2H, m), 2.11–2.12 (2H, t), 2.65–2.68 (1H, m), 4.35–4.36 (1H, d), 4.45 (2H, s), 7.20–7.21 (2H, d), 7.31–7.32 (2H, d), 7.77–7.79 (2H, d), 7.35–7.36 (2H, d), 1.85 (1H, s), 9.11 (1H, s), ^13^C NMR (DMSO, ppm) δ: 132.3(C-1), 132.5 (C-2), 127.6 (C-3), 128.0 (C-4), 128.5 (C-5), 114.6 (C-6), 159.7 (C-7), 137.1 (C-8), 126.7 (C-9), 127.0 (C-10), 132.7 (C-11), 127.1 (C-12), 126.7 (C-13), 51.1 (C-14), 169.1 (C-15), 32.5 (C-16), 23.1 (C-17), 19.6 (C-18), 14.5 (C-19), 65.4 (C-20), 167.5 (C-21), 27.1 (C-22), 16.5 (C-23), 16.5 (C-24).

AV1: IUPAC Name: 2-oxo-1,2-diphenylethyl 2-(N-((2′-(2H-tetrazol-5yl)-[1,1′-biphenyl]-4-yl)methyl)pentanamido)-3methyl butanoate yield (70%); M. P. 104 °C. Molecular formula: C_38_H_39_N_5_O_4_ and molecular weight: 629.75 gm/mol. Elemental examination (calculated) for C_38_H_39_N_5_O_4_: 72.46;H,6.22;N,11.11, (found) C,72.47;H, 6.24; N, 11.12, FT-IR ν(cm^−1^), 2959, 2881 (C−H), 1723, 1712 (C=O), 1656 (C=N), 1600 (CH=CH), 1187 (C−N), 3345, 3289 (N−H), ^1^H NMR (DMSO, ppm) δ: 0.79–0.80 (3H, t) 0.90–0.91 (6H, d), 1.23–1.25 (2H, m), 1.45–1.47 (2H, m), 2.06–2.07 (2H, t), 2.79–2.81 (1H, m), 4.43–4.44 (1H, d), 4.49 (2H, s), 7.26–7.27 (2H, d), 7.37–7.38 (2H, d), 7.67–7.67 (2H, d), 7.43–7.45 (2H, d), 7.81–7.82 (2H, d), 7.39–7.40 (2H, d), 7.55–7.58 (3H, m), 7.33–7.35 (3H, CH). ^13^C NMR (DMSO, ppm) δ: 132.1 (C-1), 132.1 (C-2), 127.6 (C-3), 128.7 (C-4), 128.5 (C-5), 114.3 (C-6), 159.3 (C-7), 137.7 (C-8), 127.9 (C-9), 127.3 (C-10), 132.9 (C-11), 127.0 (C-12), 128.7 (C-13), 57.1 (C-14), 169.0 (C-15), 36.3 (C-16), 23.5 (C-17), 19.3 (C-18), 17.5 (C-19), 65.1 (C-20), 167.1 (C-21), 27.9 (C-22), 16.7 (C-23), 16.6 (C-24), 87.3 (C-25), 132.1 (C-26), 129.3 (C-27), 128.9 (C-28), 127.9 (C-29), 128.9 (C-30), 129.3 (C-31), 178.7 (C-32), 134.5 (C-33), 128.3 (C-34), 128.1 (C-35), 131.5 (C-36), 128.1 (C-37), 128.3 (C-38).

AV2: IUPAC Name: 4-formyl-2-methoxyphenyl 2(N-((2′-(2H-tetrazole-5-yl)-[1,1′-biphenyl]-4yl)methyl)pentanamido)-3-methyl butanoate, Yield (82%); white semisolid, molecular formula: C_32_H_35_N_5_O_5_ and molecular weight: 217.27 gm/mol. Elemental examination (calculated) for C_32_H_35_N_5_O_5_: C,67.46; H,6.17;N, 12.26, (found)C,67.41; H, 6.24; N, 12.34, FT-IR ν(cm^−1^), 3289 (NH), 2959, 2870 (CH), 1712 (C=O), 1656 (C=N), 1600, 1461 (CH=CH), 1276 (CN). ^1^H NMR (DMSO, ppm) δ: 0.93–0.94 (3H, t), 0.96–0.98 (6H, d) 1.22–1.23 (2H, m), 1.55–1.57 (2H, m), 2.07–2.09 (2H, t), 2.76–2.78 (1H, m), 4.27–4.28 (1H, d), 4.57 (2H, s), 7.27–7.28 (2H, d), 7.38–7.39 (2H, d), 7.81–7.82 (2H, d), 7.43–7.45 (2H, d), 1.85 (1H, s), 9.11 (1H, s), 7.39–7.40 (1H, d), 7.63–7.64 (1H, d), 7.58 (1H, s), 3.72 (3H, s), ^13^C NMR (DMSO, ppm) δ: 132.1 (C-1), 132.7 (C-2), 127.3 (C-3), 128.9 (C-4), 129.1 (C-5), 116.1 (C-6), 159.3 (C-7), 137.7 (C-8), 127.1 (C-9), 127.5 (C-10), 133.3 (C-11), 127.5 (C-12), 126.7 (C-13), 51.4 (C-14), 169.5 (C-15), 37.1 (C-16), 23.4 (C-17), 19.3 (C-18), 27.8 (C-19), 69.7 (C-20), 167.3 (C-21), 27.5 (C-22), 21.9 (C-23), 24.5 (C-24), 142.7 (C-25), 149.7 (C-26), 109.7 (C-27), 132.7 (C-28), 124.5 (C-29), 123.1 (C-30), 166.7 (C-31), 57.9 (C-32).

AV3: IUPAC Name: 2,4,6-trinitrophenyl 2-(N-((2′-(2H-tetrazol-5-yl)-[1,1′-biphenyl]-4yl)methyl)pentanamido)-3methyl butanoate, yield (79%); green semisolid; molecular formula: C_30_H_30_N_8_O_9_ and molecular weight: 646.61 gm/mol. Elemental analysis (calculated) for C_30_H_30_N_8_O_9_: C, 55.72; H, 4.68; N, 17.33, (found) C, 55.78; H, 4.65; N, 17.30, FT-IR ν(cm^−1^), 3289 (NH), 2953, 2870 (CH), 1712, 1723 (C=O), 1656 (C=N), 1601, 1461 (C=C), 1299 (CN), ^1^H NMR (DMSO, ppm) δ: 0.75–0.77 (3H, t), 0.84–0.86 (6H, d) 1.23–1.24 (2H, m), 1.53–1.55 (2H, m), 2.19–2.21 (2H, t), 2.79–2.80 (1H, m), 4.31–4.32 (1H, d), 4.49 (2H, s). 7.25–7.26 (2H, d), 7.35–7.36 (2H, d), 7.71–7.72 (2H, d), 7.47–7.48 (2H, d), 1.81 (1H, s), 8.81 (2H, s), ^13^C NMR (DMSO, ppm) δ: 133.1 (C-1), 132.3 (C-2), 127.9 (C-3), 128.9 (C-4), 128.7 (C-5), 117.3 (C-6), 159.3 (C-7), 137.5 (C-8), 127.5 (C-9), 127.9 (C-10), 132.1 (C-11), 127.9 (C-12), 126.5 (C-13), 57.5 (C-14), 169.3 (C-15), 37.9 (C-16), 23.7 (C-17), 26.7 (C-18), 19.5 (C-19), 68.1 (C-20), 167.0 (C-21), 27.8 (C-22), 18.3 (C-23), 18.3 (C-24) 130.9 (C-25), 141.7 (C-26), 125.3 (C-27), 143.9 (C-28), 125.3 (C-29), 141.7 (C-30).

AV4: IUPAC Name: 4-aminophenyl 2-(N-((2′-(2H-tetrazol-5yl) -[1,1′-biphenyl]-4yl)methyl) pentanamido)-3methyl butanoate, yield (72%); black solid M. P. 125 °C. Molecular formula: C_30_H_34_N_6_O_3_ and molecular weight: 526.63 gm/mol. Elemental analysis (calculated) for C_30_H_34_N_6_O_3_: C, 68.41; H, 6.52; N, 15.95, (found) C, 68.45; H, 6.48; N, 15.90, FT-IR ν(cm^−1^), 3289 (N-H), 2987, 2870 (CH), 1723, (C=O), 1656 (C=N), 1589 (C=C), 1276 (CN), ^1^H NMR (DMSO, ppm) δ: 0.75–0.76 (3H, t), 0.84–0.85 (6H, d) 1.23–1.24 (2H, m), 1.37–1.38 (2H, m), 2.14–2.16 (2H, t), 2.71–2.73 (1H, m), 4.47–4.48 (1H, d), 4.41 (2H, s), 7.25–7.26 (2H, d), 7.33–7.34 (2H, d), 7.69–7.70 (2H, d), 7.39–7.40 (2H, d), 1.81 (1H, s), 6.51–6.52 (2H, d), 6.93–6.94 (2H, d), ^13^CNMR (DMSO, ppm) δ: 133.7 (C-1), 132.1 (C-2), 128.9 (C-3), 128.5 (C-4), 129.3 (C-5), 117.6 (C-6), 159.1 (C-7), 137.3 (C-8), 127.3 (C-9), 127.5 (C-10), 132.1 (C-11), 129.1 (C-12), 127.1 (C-13), 63.3 (C-14), 169.7 (C-15), 36.1 (C-16), 23.7 (C-17), 27.5 (C-18), 37.5 (C-19), 63.7 (C-20), 167.1 (C-21), 27.9 (C-22), 27.5 (C-23), 26.9 (C-24), 142.1 (C-25), 119.3 (C-26), 109.3 (C-27), 139.7 (C-28), 109.3 (C-29), 119.3 (C-30).

AV5: IUPAC Name: 2, 4-dichlorophenyl 2-(N-((2′-(2H-tetrazol-5yl)-[1,1′-biphenyl]-4 yl)methyl) pentanamido)-3methyl butanoate, yield (71%); light yellow liquid. Molecular formula: C_30_H_31_C_l2_N_5_O_3_ and molecular weight: 580.50 gm/mol. Elemental examination (calculated) for C_30_H_31_C_l2_N_5_O_3_: C,062.07; H,05.38; N,012.06, (found) C, 062.10; H, 05.40; N, 012.08, FT-IR ν(cm^−1^), 3289 (N-H), 2959, 2870 (CH), 1723 (C=O), 1656 (C=N), 1600, 1455 (C=C), 1299 (CN), ^1^H NMR (DMSO, ppm) δ: 0.81–0.82 (3H, t), 0.95–0.96 (6H, d) 1.11–1.12 (2H, m), 1.37–1.39 (2H, m), 2.02–2.04 (2H, t), 2.69–2.71 (1H, m), 4.37–4.38 (1H, d), 4.47 (2H, s), 7.27–7.28 (2H, d), 7.39–7.40 (2H, d), 7.81–7.82 (2H, d), 7.31–7.32 (2H, d), 1.77 (1H, s), 6.97–6.98 (1H, d), 7.48–7.49 (1H, d), 7.63 (1H, s), ^13^C NMR (DMSO, ppm) δ: 133.3 (C-1), 132.1 (C-2), 128.4 (C-3), 129.3 (C-4), 129.1 (C-5), 117.5 (C-6), 159.5 (C-7), 137.1 (C-8), 127.7 (C-9), 127.5 (C-10), 132.9 (C-11), 127.3 (C-12), 128.7 (C-13), 57.6 (C-14), 169.0 (C-15), 37.1 (C-16), 27.3 (C-17), 23.5 (C-18), 19.5 (C-19), 65.1 (C-20), 167.1 (C-21), 27.9(C-22), 19.7 (C-23), 19.5 (C-24), 142.3 (C-25), 130.7 (C-26), 129.9 (C-27), 143.3 (C-28), 127.9 (C-29), 123.3 (C-30).

AV6: IUPAC Name: 4-formylphenyl2-(N-((2′-(2H-tetrazol-5-yl)-[1, 1′-biphenyl]-4yl) methyl) pentanamido)-3methyl butanoate, yield (74%); white solid. M. P. 135 °C. Molecular formula: C_31_H_33_N_5_O_4_ and molecular weight: 539.62 gm/mol. Elemental analysis (calculated) for C_31_H_33_N_5_O_4_: 69.01; H, 6.61; N, 12.98; (found) C, 68.40; H, 6.70; N, 12.12, FT-IR ν(cm^−1^), 3289 (NH), 2953, 2870 (CH), 1723 (C=O), 1656 (C=N), 1601 (C=C), 1276 (CN), ^1^H NMR (DMSO, ppm) δ: 0.81–0.83 (3H, t), 0.95–0.96 (6H, d) 1.25–1.27 (2H, m), 1.51–1.53 (2H, m), 2.14–2.15 (2H, t), 2.69–2.71 (1H, m), 4.31–4.32 (1H, d), 4.39 (2H, s), 7.19–7.20 (2H, d), 7.33–7.34 (2H, d), 7.63–7.64 (2H, d), 7.39–7.40 (2H, d), 1.78 (1H, s), 6.85 (1H, s), 7.80–7.82 (2H, d), 7.92–7.94 (2H, t), ^13^C NMR (DMSO, ppm) δ: 134.0 (C-1), 133.5 (C-2), 128.1 (C-3), 128.3 (C-4), 129.1 (C-5), 117.5 (C-6), 159.5 (C-7), 137.5 (C-8), 127.6 (C-9), 128.0 (C-10), 132.5 (C-11), 127.7 (C-12), 127.8 (C-13), 58.3 (C-14), 167.6 (C-15), 37.4 (C-16), 29.3 (C-17), 23.3 (C-18), 19.5 (C-19), 65.1 (C-20), 167.3 (C-21), 27.7 (C-22), 17.3 (C-23), 16.1 (C-24), 142.5 (C-25), 169.7 (C-26), 139.7 (C-27), 124.1 (C-28), 133.1 (C-29), 133.1(C-30), 124.1 (C-31), 139.7 (C-32), 169.7 (C-33).

AV7: IUPAC Name: 3-(4-((2-(N-((2′-(2H-tetrazol-5yl)-[1,1′-biphenyl]-4 yl)methyl) pentanamido)-3methylbutanoyl)oxy)phenyl)-2amino-propanoic acid, yield (69%); light brown solid, M. P. 140 °C. Molecular formula: C_33_H_38_N_6_O_5_ and molecular weight: 598.69 gm/mol. Elemental examination, (calculated) for C_33_H_38_N_6_O_5_: C,66.20; H,06.40; N,14.04, (found) C, 66.25; H, 06.37; N, 014.09, FT-IR ν(cm^−1^), 3289 (NH), 2999, 2942 (CH), 1724 (C=O), 1657 (C=N), 1589, 1455 (CH=CH), 1299 (CN). ^1^H NMR (DMSO, ppm) δ: 0.75–0.76 (3H, t), 0.86–0.87 (6H, d) 1.28–1.30 (2H, m), 1.47–1.49 (2H, m), 2.07–2.09 (2H, t), 2.72–2.73 (1H, m), 4.31–4.32 (1H, d), 4.52 (2H, s), 7.27–7.28 (2H, d), 7.37–7.38 (2H, d), 7.71–7.72 (2H, d), 7.33–7.34 (2H, d), 1.71 (1H, s), 7.18–7.19 (2H, d), 7.22–7.23 (2H, d), 3.17–3.18 (2H, d), 4.11–4.12 (1H, t), 4.85 (2H, s), 8.99 (1H, s), ^13^C NMR (DMSO, ppm) δ: 133.6 (C-1), 132.4 (C-2), 128.7 (C-3), 128.3 (C-4), 128.1 (C-5), 117.1 (C-6), 159.5 (C-7), 136.3 (C-8), 127.5 (C-9), 127.5 (C-10), 132.9 (C-11), 127.9 (C-12), 127.8 (C-13), 57.9 (C-14), 167.1 (C-15), 37.5 (C-16), 23.9 (C-17), 24.5 (C-18), 26.7 (C-19), 65.1 (C-20), 165.3 (C-21), 27.7 (C-22), 17.5 (C-23), 17.5 (C-24), 144.7 (C-25), 119.7 (C-26), 128.7 (C-27), 131.7 (C-28), 128.7 (C-29), 119.7 (C-30), 35.7 (C-31), 59.5 (C-32), 169.7 (C-33).

AV8: IUPAC Name: 2-isopropyl-5-methylphenyl 2-(N-((2′-(2H-tetrazol-5yl)-[1, 1′-biphenyl]-4 yl)methyl)pentanamido)-3methyl butanoate, yield (75%); pink semisolid. Molecular formula: C_34_H_41_N_5_O_3_ and molecular weight: 567.72 gm/mol. Elemental analysis (calculated) for C_34_H_41_N_5_O_3_: C, 71.93; H, 7.27; N, 12.34; (found) C, 71.90; H, 7.22; N, 12.40, FT-IR ν(cm^−1^), 3289 (NH), 2987 (CH), 1712 (C=O), 1656 (C=N), 1612 (C=C), 1299 (CN), ^1^H NMR (DMSO, ppm) δ: 0.77–0.78 (3H, t), 0.88–0.89 (6H, d) 1.20–1.21 (2H, m), 1.37–1.39 (2H, m), 2.01–2.03 (2H, t), 2.73–2.74 (1H, m), 4.26–4.27 (1H, d), 4.37 (2H, s), 7.25–7.27 (2H, d), 7.36–7.37 (2H, d), 7.63–7.64 (2H, d), 7.38–7.39 (2H, d), 1.77 (1H, s), 6.94 (1H, s), 6.88–6.89 (1H, d), 7.12–7.13 (1H, d), 2.27 (3H, s), 2.91–2.93 (1H, m), 1.12 (6H, d), ^13^C NMR (DMSO, ppm) δ: 134.3 (C-1), 132.7 (C-2), 129.7 (C-3), 128.9 (C-4), 128.1 (C-5), 117.3 (C-6), 157.1 (C-7), 137.5 (C-8), 127.3 (C-9), 127.0 (C-10), 132.1 (C-11), 127.3 (C-12), 126.5 (C-13), 57.5 (C-14), 167.3 (C-15), 37.5 (C-16), 23.9 (C-17), 22.3 (C-18), 16.5 (C-19), 65.7 (C-20), 167.5 (C-21), 27.9 (C-22), 19.5 (C-23), 19.8 (C-24) 144.3 (C-25), 139.9 (C-26), 123.7 (C-27), 124.5 (C-28), 135.7 (C-29), 120.5 (C-30), 15.7 (C-31), 29.5 (C-32), 19.7 (C-33,C-34).

AV9: IUPAC Name: naphthalen-1-yl 2-(N-((2′-(2H-tetrazol-5-yl)-[1,1′-biphenyl]-4 yl)methyl) pentanamido)-3methyl butanoate, yield (72%); brown Semisolid, Molecular formula: C_34_H_35_N_5_O_3_ and molecular weight: 561.67 gm/mol. Elemental analysis (calculated) for C_34_H_35_N_5_O_3_: C, 72.70; H, 6.28; N, 12.47, (found) C, 72.65; H, 6.25; N, 12.42, FT-IR ν(cm^−1^), 3289 (NH), 2959, 2942 (CH), 1712 (C=O), 1657 (C=N), 1612 (C=C), 1276 (C−N), ^1^H NMR (DMSO, ppm) δ: 0.76–0.78 (3H, t), 0.95–0.96 (6H, d) 1.25–1.26 (2H, m), 1.55–1.57 (2H, m), 2.05–2.07 (2H, t), 2.77–2.79 (1H, m), 4.44–4.45 (1H, d), 4.57 (2H, s), 7.25–7.26 (2H, d), 7.37–7.38 (2H, d), 7.63–7.64 (2H, d), 7.39–7.40 (2H, d), 1.75 (1H, s), 7.92–7.93 (2H, d), 7.55–7.57 (2H, t), 7.46–7.47 (1H, t), 7.05–7.06 (1H, t), 7.14–7.15 (1H, t), ^13^C NMR (DMSO, ppm) δ: 134.1 (C-1), 132.8 (C-2), 129.2 (C-3), 128.9 (C-4), 128.7 (C-5), 116.6 (C-6), 159.1 (C-7), 137.8 (C-8), 127.2 (C-9), 127.5 (C-10), 132.6 (C-11), 127.9 (C-12), 126.9 (C-13), 57.4 (C-14), 169.7 (C-15), 35.1 (C-16), 24.3 (C-17), 20.5 (C-18), 15.7 (C-19), 65.9 (C-20), 167.3 (C-21), 27.7 (C-22), 19.5 (C-23), 19.5 (C-24), 144.3 (C-25), 129.9 (C-26), 123.7 (C-27), 128.1 (C-28), 135.7 (C-29), 120.5 (C-30), 15.7 (C-31), 29.5 (C-32), 125.7 (C-33), 115.3 (C-34).

AV10: IUPAC Name: naphthalen-2-yl 2-(N-((2′-(2H-tetrazol-5-yl)-[1,1′-biphenyl]-4 yl)methyl)pentanamido)-3methyl butanoate, yield (67%); brown semisolid, Molecular formula: C_34_H_35_N_5_O_3_ and molecular weight: 561.67 gm/mol. Elemental analysis (calculated) for C_34_H_35_N_5_O_3_: C 72.70; H, 6.28; N, 12.47, (found) C, 72.65; H, 6.24; N, 12.43, FT-IR ν (cm^−1^), 3295 (NH), 2953 (CH), 1656 (C=N), 1612 (C=C), 1276 (CN), ^1^H NMR (DMSO, ppm) δ: 0.88–0.89 (3H, t), 0.94–0.95 (6H, d) 1.17–1.20 (2H, m), 1.41–1.44 (2H, m), 2.11–2.13 (2H, t), 2.69–2.72 (1H, m), 4.39–4.42 (1H, d), 4.61 (2H, s), 7.23–7.25 (2H, d), 7.31–7.34 (2H, d), 7.66–7.68 (2H, d), 7.37–7.39 (2H, d), 1.95 (1H, s), 7.89 (1H, s), 7.45–7.46 (1H, d), 7.97–7.98 (1H, d), 8.05–8.07 (2H, m), 7.58–7.59 (1H, t), 7.68–7.69 (1H, t), ^13^C NMR (DMSO, ppm) δ: 132.8 (C-1), 132.6 (C-2), 130.1 (C-3), 129.7 (C-4), 129.0 (C-5), 114.2 (C-6), 157.3 (C-7), 137.5 (C-8), 127.3 (C-9), 127.0 (C-10), 132.3 (C-11), 127.4 (C-12), 127.5 (C-13), 59.3 (C-14), 166.9 (C-15), 37.9 (C-16), 23.7 (C-17), 19.8 (C-18), 17.5 (C-19), 65.9 (C-20), 167.1 (C-21), 27.9 (C-22), 16.4 (C-23), 16.4 (C-24), 144.0 (C-25), 139.7 (C-26), 123.1 (C-27), 124.1 (C-28), 135.1 (C-29), 120.3 (C-30), 123.7 (C-31), 129.5 (C-32), 119.7 (C-33), 132.1 (C-34).

AV11: IUPAC Name: 2-((2-(N-((2′-(2H-tetrazole-5-yl)-[1,1′-biphenyl]-4yl) methyl) pentanamido)-3-methylbutanoyl) oxy) benzoic acid, yield (70%); semisolid. Molecular formula: C_31_H_33_N_5_O_5_ and molecular weight: 555.62 gm/mol. Elemental examination (calculated) for C_31_H_33_N_5_O_5_: C, 67.01; H, 05.99; N, 012.60, (found)C, 67.05; H,05.93; N,012.54, FT-IR ν (cm^−1^), 3289 (NH), 2959 (CH), 1617 (C=N), 1601 (CH=CH), 1299 (CN), 1723 (C=O), ^1^H NMR (DMSO, ppm) δ: 0.86–0.87 (3H, t), 0.92–0.93 (6H, d) 1.20–1.21 (2H, m), 1.41–1.42 (2H, m), 2.15–2.17 (2H, t), 2.69–2.71 (1H, m), 4.35–4.36 (1H, d), 4.57 (2H, s), 7.22–7.24 (2H, d), 7.35–7.36 (2H, d), 7.77–7.78 (2H, d), 7.41–7.42 (2H, d), 1.89 (1H, s), 9.10 (1H, s), 7.81–7.82 (1H, d), 8.01–8.02 (2H, m), 7.71–7.72 (1H, t), ^13^C NMR (DMSO, ppm) δ: 132.0 (C-1), 132.0 (C-2), 127.8 (C-3), 129.2 (C-4), 128.8 (C-5), 114.7 (C-6), 159.8 (C-7), 137.4 (C-8), 126.9 (C-9), 127.1 (C-10), 132.9 (C-11), 127.8 (C-12), 126.9 (C-13), 59.1 (C-14), 169.0 (C-15), 39.1 (C-16), 23.9 (C-17), 19.9 (C-18), 14.7 (C-19), 65.3 (C-20), 167.7 (C-21), 27.9 (C-22), 16.9 (C-23), 16.1 (C-24), 144.3 (C-25), 139.9 (C-26), 123.7 (C-27), 124.5 (C-28), 135.7 (C-29), 120.5 (C-30), 155.7 (C-31).

### 3.3. ADME Studies

All of the drug’s derivatives, and pharmacokinetic parameters (absorption, distribution, metabolism, and excretion, or ADME) were computed using the swissADME web server [28]. To predict the pharmacokinetic studies, the smilies format for each produced chemical was applied.

### 3.4. Free Radical Scavenging Activity

The DPPH assay was carried out to estimate the antioxidant potential among all the compounds [29]. Then, 2 mL (0.2 mg/mL in methyl alcohol) of DPPH was mixed with 1.0 mL (0.5 mg/mL in methyl alcohol) of the sample solution in a test tube. All the reaction tubes were covered with aluminum foil and kept aside for 30 min to complete the reaction. The activity was carried out in a dark room. The shift in the hue of the solution from purple to yellow indicates that the DPPH radical has been scavenged. A UV-visible spectrophotometer was employed to observe a reduction in the concentration of free radicals at 517 nm. The absorbance of the control (1.0 mL methanol + 2 mL DPPH solution) was also measured. All samples were evaluated in a triplicate manner, and the mean results were recorded. The standard antioxidant (ascorbic acid) was evaluated and set as a reference/standard. The percentage of inhibition (%) was calculated using the following formula in terms of scavenging activity(S):S (%) = [(Ab − Aa)/Ab] × 100(1)
where Ab = Absorbance of control; and Aa = Absorbance of the test sample

### 3.5. Anti-Hypertensive Activity

Cervical disruption was used to eliminate male Wistar rats weighing 200 g and 300 g. The thoracic aorta segments (3 mm to 5 mm) were slightly pressed and endothelium was removed. The tissues were placed in an organ bath filled with 10 mL of Krebs-Henseleit solution and kept at 37 °C. Then 95% oxygen and 5% carbon dioxide were passed through the above-mentioned solution. To record organ responses to phenylephrine, (10^−6^ M) (Sigma, St. Louis, MO, USA) simulated contractions; isometric transducers and Gemini recorders (Ugo Basile, Varese, Italy) were assembled. Aortic rings were recon traced with phenylephrine (10^−6^ M) and subjected to acetylcholine (10^−6^ M) for recording endothelium-dependent relaxations (EDR). All synthesized compounds along with the standard (valsartan) were mixed in DMSO; whereas synthesized derivatives of concentration 10^−4^ M were added in an organ bath. In conscious rats, systolic blood pressure was recorded by applying the tail-cuff method. Wistar rats of 200 g to 250 g weight were selected and divided into groups of five individuals. The chemicals were injected intraperitoneally at a dosage of 80 mg/kg body weight. The tail-cuff and a piezoelectric pulse sensor were coupled to a blood pressure analyzer and were positioned at the base of the tail (May 9610, Ankara, Turkey) [24].

All results were presented as the mean of SEM. (n = 5) and test compounds were employed for statistical significance determination.

### 3.6. In Vitro Urease Enzyme Inhibition

The activity performed was an improved procedure of the Berthelot assay. The mixture was prepared using 10 µL phosphate buffer (pH 7.0), sample solution 10 µL, and enzyme solution 25 µL (0.135 units). For 5 min, the mixture was incubated at 37 °C. Each well in a 96-well plate received 40 µL of urea stock solution (20 mM); incubated for 10 min at 37 °C. Afterwards, 115 µL phenol hypochlorite (45 µL phenol + 70 µL alkali) per well was added. Furthermore, all the contents were incubated for 10 min at 37 °C. Absorbance at 625 nm was measured [30]. Using the provided formula, the % of enzyme-inhibition and IC-50 values were computed
Inhibition (%age) = [Control − test/100(control)] 
where Control = [Total enzyme activity minus inhibitor]. The test is the test compounds’ activity

### 3.7. Antimicrobial Screening

Antibacterial activity of 2-(N-((2′-(2H-tetrazole-5-yl)-[1,1′-biphenyl]-4yl)-methyl)-pentanamido)-3-methyl butanoic acid and derivatives was evaluated using the agar-well diffusion method against four strains of bacteria *Bacillus pumilus*, *Escherichia coli*, *Staphylococcus aureus*, and *Pseudomonas aeruginosa*. After sterilizing the Mueller Hinton agar medium on plates in an autoclave and allowing it to cool to roughly 45 °C, a 100 µL standard inoculum was aseptically mixed, chilled till 37 °C, and 6 mm wells were formed in the inoculated plates. The volume of 100 μL (100 mg/5 mL in DMSO), ceftriaxone sodium (20 mg/10 mL in DMSO) as a positive control, and pure DMSO as a negative control; 30 μL of the sample solution was incorporated into these wells. These primed plates were incubated at 37 °C for 24 h prior to record inhibitory zones in millimeters [31].

### 3.8. Molecular Docking

The drug’s binding mechanism at the active site of a chosen enzyme (urease inhibitor) was revealed using the Glide in-silico procedure (Schrödinger software program Maestro 2017–2 (Glide, Schrödinger, LLC, New York, NY, USA, 2017)). The PDB ID 1E9Y was copied from the protein data bank and used for docking studies. (www.rcsb.org, accessed on 16 November 2022).

#### 3.8.1. Ligand Preparation

Every ligand was haggard implementing Maestro software and organized for docking using the Lig-Prep tool (Schrödinger, LLC, New York, NY, USA, 2017). In silico studies for docking were carried out in Schrodinger Software utilizing the Glide (Grid-based Ligand Docking with Energetics) database (maestro) [32].

#### 3.8.2. Protein Preparation

The clear structure of urease (PDB id: 1E9Y) was taken into consideration to catch perceptions for the binding means of ligands. The X-ray structure of the protein was constructed via Maestro 11.4′s Protein Preparation Wizard [33] following the steps (i) Schrodinger’s Prime 3.0 was used to add side chains that were missing into the structure of the enzyme. (ii) Hydrogen-bonded atoms were incorporated; water molecules that were distant from 5 of the co-crystallized ligands were eliminated (iii) Using Epik of the complete system, protonation states managed to a pH 7.0 ± 2.0. (iv) Hydrogen bonding ties, flip-orientations/tautomeric forms of Gln, Asn, and HIS remains were adjusted. (v) The geometric optimization was achieved using the OPLS force field up to the specified 0.3 Å root mean square deviation (RMSD) [34].

#### 3.8.3. Receptor-Grid Generation

Furthermore; by applying the receptor-grid generation component in the Glide database, a protein net was produced. Using a co-crystal ligand, the binding sites were defined.

#### 3.8.4. Docking Studies

The molecular-docking studies were executed through the Extra-precision (XP) method via Glide [35,36]. It also facilitates favorable ligand positions for further scrutinizing active sites for the ligand attachments. The docking results offered the best postures besides the docking score and glide score. The outcomes of the docking studies and active derivatives are stated concisely in Table 7 [37].

### 3.9. Statistical Analysis

Biological activity values are presented as mean or mean + SEM. Student’s *t*-test was used to establish statistical significance, and a level of significance of *p* < 0.05 was used.

## 4. Conclusions

This study was conducted to synthesize and characterize the novel ester derivatives hoping the discovery of new structural moieties could serve as a lead in the treatment of hypertension along with its associated pathological conditions. Newly synthesized derivatives were subjected to evaluate their antihypertensive and antioxidants potentials along with urease inhibition activity, molecular docking, and ADME studies. The spectral studies (FTIR and NMR) confirmed the synthesis of the derivatives. Biological assays revealed spectacular activity against hypertension and urease inhibition. The compound AV9 exhibited higher gastrointestinal absorption comparable with the parent drug. In addition, AV1-AV11 showed excellent antioxidant potentials except AV7. Moreover, the AV2 compound showed the maximum activity against hypertension. Furthermore, the compound AV2 indicated remarkable urease inhibition, even more than thiourea used as standard while other compounds also exhibited inhibition potential against urease. So, it can be suggested that these synthesized compounds can lead to further drug development, and their safety and efficacy profiles are recommended to be investigated in other experimental models.

## Data Availability

This article contains all the information which was analyzed and calculated as a part of this study. Further inquiries can be entertained by the corresponding author.

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
