# Peer review of "Synthesis, Characterization, and Biological Evaluation of 2-(N-((2′-(2H-tetrazole-5-yl)-[1,1′-biphenyl]-4yl)-methyl)-pentanamido)-3-methyl Butanoic Acid Derivatives"

_molecules, 2023, doi:10.3390/molecules28041908_

Round 1

Reviewer 1 Report

In this manuscript Anum Masood et al. describe a preparation of the library of valsartan [2-(N-((2'-(2H-tetrazole-5-yl)-[1,1'-biphenyl]-4-yl)-methyl)pentanamido)-3-methylbutanoic acid] esters that show promising activity against hypertension, and urease inhibition, as well as antioxidant potentials. Overall, the manuscript would be interesting for the readers because it produces a good addition to the chemical knowledge in this field. However, the manuscript needs to be very carefully checked and revised before it can be accepted for the publishing in the journal Molecules. A major revision is required.

Manuscript comments:

1.    All compounds’ names should be checked and corrected if necessary.

2.    L10: Is it Universiti of Teknologi or Universiti Teknologi?

3.    The study compounds should be included in the Keywords.

4.    For the citation throughout the manuscript, please follow the Guidelines for the authors ([ ] instead of ( )).

5.    The characterization of ester structure would be removed from the Introduction.

6.     All Latin names should be italicized.

7.    Sections 2 and 3 should be combined to the section named Results and discussion, where the obtained data should be discussed. All tables should be should be inserted into the main text close to their first citation. All spectral data should be moved to the Materials and methods section. As an example, for the preparation of the article, please select any article published in Molecules that discusses synthesis and bioevaluation.

8.    In the description of the spectra (for all 1H NMR spectra), please follow any article published in Molecules that contains the experimental section.

9.    Please check all IUPAC names of AV1-11 compounds in the Results section and correct if necessary.

10. It is difficult to understand such characterization of C atoms (C-001, C-002 etc) in the description of the 13C NMR spectra. Car, CH3, CH2CH2 etc is better choice.

11. Figures should be renumbered and cited exactly in the text.

12. Please check and correct the values and statements in the Discussion section according to the Tables data.

13. L359, In the Preparation of esters section, the used amounts (in g, mL and mol or mmol) of the reactants and solvents should be included.

14. L367: Figure 1. Synthesis of Valsartan Derivatives AV1-AV11 should be renamed as Scheme 1. Synthesis of Valsartan Derivatives AV1-AV11.

15. Under the scheme title, Reagents and reaction conditions should be included. They should be moved here from the scheme: Reagents and conditions: a reagent, solvent, catalyst, reaction temperature, time.

16. The scheme shows the usage of sulphuric acid, however in 4.2 Preparation of esters (L360-364) it is missing.

17. mL instead of ml.

18.  In the manuscript, the values and units would be written in the same way (with the spacing between the value and unit or without the spacing, but in the same way throughout all manuscript).

19. Conclusion should be improved.

20. References list should be checked and corrected according the Guidelines.

21. There are many typo errors and inaccuracies in the manuscript.

Author Response

Response to reviewer 1

Manuscript comments:

  1. All compounds’ names should be checked and corrected if necessary.

All compunds names are checked and corrected where required.

  1. L10: Is it Universiti of Teknologi or Universiti Teknologi?

It is “ Universiti Teknologi”.

  1. The study compounds should be included in the Keywords.

The studied compound is included in the keywords.

  1. For the citation throughout the manuscript, please follow the Guidelines for the authors ([ ] instead of ( )).

Reference guidelines have been followed throughout the manuscript.

  1. The characterization of ester structure would be removed from the Introduction.

The characterization of ester structure removed from the Introduction.

  1. All Latin names should be italicized.

All latin names are italicized

  1. Sections 2 and 3 should be combined to the section named Results and discussion, where the obtained data should be discussed. All tables should be should be inserted into the main text close to their first citation. All spectral data should be moved to the Materials and methods section. As an example, for the preparation of the article, please select any article published in Molecules that discusses synthesis and bioevaluation.

Section 2 and 3 have been combined under heading “ Results and Discussion” and all tables are inserted where they are cited first in the main text. All the spectral has been moved to  section 3 “Material and method”.

  1. In the description of the spectra (for all 1H NMR spectra), please follow any article published in Molecules that contains the experimental section.

All 1H NMR spectras are arranged according to the journal requirments.

  1. Please check all IUPAC names of AV1-11 compounds in the Results section and correct if necessary.

           All IUPAC names are double checked and corrected where necessary

  1. It is difficult to understand such characterization of C atoms (C-001, C-002 etc) in the description of the 13C NMR spectra. Car, CH3, CH2CH2etc is better choice.

         In characterization, the C-001, C002 etc are changed to C-1, C-2, C-3 and so on for better    understanding.

  1. Figures should be renumbered and cited exactly in the text.

       Figures are renumbered and cited exactly in the main text.

  1. Please check and correct the values and statements in the Discussion section according to the Tables data.

       All the statements are checked and corrected according to tables in discussion section.

  1. L359, In the Preparation of esters section, the used amounts (in g, mL and mol or mmol) of the reactants and solvents should be included.

                Preparation of ester section has been changed to “General procedure for the synthesis of valsartan series AV1-AV11” and the used amounts of reactants and solvents are included.

  1. L367: Figure 1. Synthesis of Valsartan Derivatives AV1-AV11 should be renamed as Scheme 1.Synthesis of Valsartan Derivatives AV1-AV11.

        Figure 1. has been changed under the name “Scheme 1. Synthesis of valsartan derivatives AV1-AV11”.

  1. Under the scheme title, Reagents and reaction conditionsshould be included. They should be moved here from the scheme: Reagents and conditions: a reagent, solvent, catalyst, reaction temperature, time.

       Under the scheme title. Reagents and reaction conditions are indicated.

  1. The scheme shows the usage of sulphuric acid, however in 4.2 Preparation of esters (L360-364) it is missing.

        The scheme shows the usage of sulphuric acid as a catalyst and it is added in 4.2.1 General procedure for the synthesis of Valsartan series AV1-AV11.

  1. mL instead of ml.

       ml is corrected as mL.

  1. In the manuscript, the values and units would be written in the same way (with the spacing between the value and unit or without the spacing, but in the same way throughout all manuscript).

      All the values and units are written in same way throughout the text now without spacing.

  1. Conclusion should be improved.

     Conclusion has been improved.

  1. References list should be checked and corrected according the Guidelines.

     References are corrected according the guidelines of the journal.

  1. There are many typo errors and inaccuracies in the manuscript.

    All the errors are checked and corrected accordingly.

Reviewer 2 Report

This paper is interesting and need only of minor revision.

It report the synthesis and characterization of the ester derivatives produced from the compound (S)-3-methyl-2-(N-[2’-(2H-1,2,3,4yl] pentanamido) butanoic acid. The esters, probably because higher lipophilicity than acid precursor, showed interesting activity against hypertension, and urease inhibition and antioxidant potentials.

Minor revision:

1) Figures 3 and 4 need to be improved, they are difficult to read.

2) Figure 1 appears in the text after Figure 4, rearrange them.

3)Figure 1 does not show the reaction solvent, but it is written that the reaction is carried out under reflux. Then concentrated H2SO4 appears and this certainly creates confusion, it is advisable to fix this figure with the reflux solvent and the right reagents.

4) mass numbers for 1H and 13C must be superscripts

Author Response

Response to reviewer 2

Manuscript comments:

  • Figures 3 and 4 need to be improved, they are difficult to read.

Figure 3 and 4 are replaced with new figures and rearranged as Figure 1 and Figure 2 according to numbering in the manuscript as figure numbers are reaaranged.

  • Figure 1 appears in the text after Figure 4, rearrange them.

Figure numbers are rearranged throughout the manuscript.

  • Figure 1 does not show the reaction solvent, but it is written that the reaction is carried out under reflux. Then concentrated H2SO4 appears and this certainly creates confusion, it is advisable to fix this figure with the reflux solvent and the right reagents.

Reaction is carried using methanol as a solvent and conc. Sulphuric acid used as a catalyst and it is mentioned now under the general procedure for the synthesis of valsartan series AV1-AV11.

      4) mass numbers for 1H and 13C must be superscripts

            Mass numbers are changed to superscripts as 1H and 13C.

Reviewer 3 Report

The article represents interesting research, but from a chemical point of view, this is a too simple modification, and I would recommend supplementing with other types of alcohols and making a SAR study as well. However, the article has sufficient novelty and interest for readers, and I recommend publishing it after a minor revision. Also, the article has technical flaws throughout that create the impression of careless design (superscripts and subscripts, missed italic mode, missed spaces, etc.).

Here are some points that authors should take into account:

1. In my opinion, for better understanding, the authors would do well to provide a scheme of synthesis of new compounds, and an extended description of this synthesis in part 2 (results). All spectral data should be transferred to Section 4.2 of Materials and Methods. All tables should be located after they were mentioned (i.e., through the Part 3 Discussion).

2. There should be a space before the reference number (for example, line 30). Please double-check the paper.

3. In vitro and in vivo should be italicized without a hyphen. Please double-check all paper (for example, line 50)

4. There is no information about what AVO actually means, although it is mentioned in every table. I guess it is valsartan, but it must be noticed in the main text anyway.

5. Some grammar and punctuation errors

Line 55, after (1926) should be a comma

line 57 Should be "the same"

Line 68  Should be «have intensified to search for…»

Line 70 «has shown a close»

Line 72 impacts change to play

Line 276 fix 1H and 13C

Line 299, 304, 306, 309, 477 missed comma before as, while and which

Line 367 Should be "Scheme. Synthesis of Valsartan Derivatives AV1-AV11"

Line 477 Should be «antioxidant»

Please proofread all text for grammar.

6. I would see more detailed type of alcohols (represent in the form of structural formulas) and it would be nice if authors will add some SAR results about influence of used alcohols in discussion section (at least about urease enzyme inhibition) 

Author Response

Response to reviewer 3

  1. In my opinion, for better understanding, the authors would do well to provide a scheme of synthesis of new compounds, and an extended description of this synthesis in part 2 (results). All spectral data should be transferred to Section 4.2 of Materials and Methods. All tables should be located after they were mentioned (i.e., through the Part 3 Discussion).

General procedure for the synthesis of ester has been added and all the spectral data has been transferred from results to material and methods. All the tables are transferred in results and discussion where they are mentioned in the text.

  1. There should be a space before the reference number (for example, line 30). Please double-check the paper.

Space is added before the reference number and it is checked throughtout the manuscript.

  1. In vitroand in vivo should be italicized without a hyphen. Please double-check all paper (for example, line 50)

In vitro and in vivo are italicized without a hyphen and it is checked throughout the paper.

  1. There is no information about what AVO actually means, although it is mentioned in every table. I guess it is valsartan, but it must be noticed in the main text anyway.

Valsartan is coded AV0 and word “parent molecule” is used for valsartan and clearly mentioned in the text now.

  1. Some grammar and punctuation errors

Line 55, after (1926) should be a comma

Comma is added after (1926).

line 57 Should be "the same"

the word “same” is corrected as “the same”

Line 68  Should be «have intensified to search for…»

Line 68 has been fixed.

Line 70 «has shown a close»

“has shown close” changed to “has shown a close”.

Line 72 impacts change to play

This point has been fixed.

Line 276 fix 1H and 13C

1H and 13C changed to 1H and 13C.

Line 299, 304, 306, 309, 477 missed comma before as, while and which

These points are addressed.

Line 367 Should be "Scheme. Synthesis of Valsartan Derivatives AV1-AV11"

This point is addressed accordingly

Line 477 Should be «antioxidant»

Antioxidants changed to antioxidant

Please proofread all text for grammar.

Whole manuscript has been read carefully and checked for grammar mistakes.

  1. I would see more detailed type of alcohols (represent in the form of structural formulas) and it would be nice if authors will add some SAR results about influence of used alcohols in discussion section (at least about urease enzyme inhibition) 

Alcoholic and phenolic reagents are represented with structural formulas. Some SAR results are added for urease enzyme inhibition in discussion section.

Round 2

Reviewer 1 Report

Manuscript comments:

a lot has been corrected, but quite a lot of errors and inaccuracies have been left:

1. All names of compounds (in the entire text), as well as in the experimental part, should be corrected;

2. Naming and citations of tables, figures and diagrams not according to instructions;

3. The line presented in L122 does not really correspond to the data in Table 5.

4. Is the molar mass measured in gm/mol (throughout the experimental part, L226, 239, etc.).

5. Inserted a cluttered picture of the alcohols/phenols used (Figure 3, L218, 219), whereas under Scheme 1 L216 there is a table with useless crude formulas, and in their place I have shown the R structures of the alcohols/phenols.

6. It is written that all compounds are recrystallized from ethyl alcohol. So where does semisolid come from?

Author Response

Response to Reviewer 1 Comments

  1. All names of compounds (in the entire text), as well as in the experimental part, should be corrected;

All names of compounds are corrected throughout the text and experimental section.

  1. Naming and citations of tables, figures and diagrams not according to instructions;

Naming and citations of tables are corrected according to the instructions.

  1. The line presented in L122 does not really correspond to the data in Table 5.

This line is rearranged according to the table 5.

  1. Is the molar mass measured in gm/mol (throughout the experimental part, L226, 239, etc.).

 These are the molecular weights found for newly synthesized compounds that are presented in g/mol.

  1. Inserted a cluttered picture of the alcohols/phenols used (Figure 3, L218, 219), whereas under Scheme 1 L216 there is a table with useless crude formulas, and in their place I have shown the R structures of the alcohols/phenols.

This point addressed according to the comment

  1. It is written that all compounds are recrystallized from ethyl alcohol. So where does semisolid come from?

  We have recrystallized only those compounds which were obtained in solid form (not followed for semisolid).